# Fully Automatic Neural Network Reduction for Formal Verification

**Tobias Ladner**                                                                *tobias.ladner@tum.de*
**Matthias Althoff**                                                                     *althoff@tum.de*
*School of Computation, Information and Technology*
*Technical University of Munich, Germany*

**Reviewed on OpenReview:** *https://openreview.net/forum?id=gmflcWlVMl*

## Abstract

Formal verification of neural networks is essential before their deployment in safety-critical applications. However, existing methods for formally verifying neural networks are not yet scalable enough to handle practical problems under strict time constraints. We address this challenge by introducing a fully automatic and sound reduction of neural networks using reachability analysis. The soundness ensures that the verification of the reduced network entails the verification of the original network. Our sound reduction approach is applicable to neural networks with any type of element-wise activation function, such as ReLU, sigmoid, and tanh. The network reduction is computed on the fly while simultaneously verifying the original network and its specification. All parameters are automatically tuned to minimize the network size without compromising verifiability. We further show the applicability of our approach to convolutional neural networks by explicitly exploiting similar neighboring pixels. Our evaluation shows that our approach reduces large neural networks to a fraction of the original number of neurons and thus shortens the verification time to a similar degree.

## 1 Introduction

Neural networks achieve impressive results in a variety of fields, including natural language processing (Nassif et al., 2019), computer vision (Li et al., 2022), and medical imaging (Karimi & Salcudean, 2019). In recent years, neural networks have been deployed in safety-critical environments, such as human-robot interaction (Mukherjee et al., 2022) and autonomous driving (Zablocki et al., 2022). As real-life applications are inherently exposed to noise, such as measurement inaccuracies and external disturbances, the deployment of neural networks in safety-critical environments is limited due to their sensitivity to adversarial attacks (Goodfellow et al., 2015): Even small perturbations of the input to a neural network, which are often barely noticeable to the human eye, can lead to unexpected outputs, e.g., a different predicted classification of an image or a controller returning an unsafe action. Thus, the formal verification of neural networks has gained importance in recent years (Brix et al., 2024; Lopez et al., 2024), where approaches rigorously prove that the output of neural networks meets given specifications.

### 1.1 Related Work

**State of the art.**     Formally verifying neural networks is computationally expensive, e.g., verifying a neural network with ReLU activations is NP-hard (Katz et al., 2017). Thus, instead of applying complete algorithms (Huang et al., 2017; Katz et al., 2017), recent developments are made towards incomplete algorithms, where the neural networks are abstracted to enclose the exact behavior of the network. These approaches often formulate the formal verification of neural networks as an optimization problem (Katz et al., 2017; Zhang et al., 2018; Katz et al., 2019; Müller et al., 2022) or use reachability analysis (Singh et al., 2018a; Bogomolov et al., 2019; Bak, 2021; Lopez et al., 2023b; Kochdumper et al., 2023; Ladner & Althoff, 2023).

Optimization-based verifiers reason about neural networks by introducing relaxed, linear constraints for the activation functions and solving these relaxed problems using linear programming, satisfiability modulo theories (SMT) solvers (Katz et al., 2017; Zhang et al., 2018; Katz et al., 2019; Müller et al., 2022), or symbolic interval propagation (Henriksen & Lomuscio, 2020; Singh et al., 2019; Brix & Noll, 2020). Branch-and-bound strategies (Bunel et al., 2020) can be beneficial by splitting the problem at the neuron level (Botoeva et al., 2020; Singh et al., 2018b), e.g., by splitting ReLU neurons into their linear parts. In general, algorithms that split the problem lead to an exponential time complexity (Katz et al., 2017), so that current state-of-the-art tools (Brix et al., 2024) use more advanced strategies (Wang et al., 2021; Ferrari et al., 2022; Shi et al., 2023).

Verifiers using reachability analysis propagate sets through the neural network and verify a given specification using the computed output set. Simple representatives of this approach use pure interval arithmetic (Pulina & Tacchella, 2010) or convex set representations such as zonotopes (Gehr et al., 2018; Singh et al., 2018a). As with optimization-based verifiers, splitting the set can improve the results (Xiang et al., 2018; Kopetzki & Günnemann, 2021). Non-convex set representations are used to tightly enclose the output due to the inherent nonlinearity of neural networks, including Taylor models (Ivanov et al., 2021; Bogomolov et al., 2019; Huang et al., 2022), star sets (Bak, 2021; Lopez et al., 2023b), and polynomial zonotopes (Kochdumper et al., 2023; Ladner & Althoff, 2023). However, the scalability to state-of-the-art networks remains a major challenge for optimization-based approaches and approaches based on reachability analysis (Brix et al., 2024).

**Challenges.** Safety-critical environments are often time-critical, e.g., verifying a controller of an autonomous car (Lopez et al., 2024), or require verifying many instances, e.g., in video verification and in formal explainable artificial intelligence (Bassan & Katz, 2023). Thus, each verification query should be solved quickly to reduce the overall verification time. However, many state-of-the-art verifiers focus on very challenging queries, with large timeouts per query. For example, the allowed time to verify a *single* query in the last VNN-COMP (Brix et al., 2024) range up to 20 minutes, which is not feasible in time-critical scenarios.

**Sound neural network reduction.** One promising research direction to tackle this issue is sound neural network reduction (Boudardara et al., 2023c): Sound neural network reduction reduces the number of neurons and provides formal bounds for the maximum error due to this reduction. Then, we can reason about the original network by verifying the reduced network, which usually requires less computation time. This research direction is closely related to neural network compression (Zhangheng et al., 2022; Deng et al., 2020), where the main goal is to reduce memory usage and computation time, e.g., for deployment in embedded systems (Deng et al., 2020). This can be done as neural networks are typically over-parametrized (Neyshabur et al., 2018). Examples of compression techniques are quantization (Cheng et al., 2017) and pruning (Gonzalez-Carabarin et al., 2024). However, the lack of formal error bounds prevents applying these techniques to the formal verification of neural networks.

Several approaches to soundly reduce neural networks with formal error bounds emerged in recent years: An early approach categorizes neurons based on analytic properties and merges neurons of the same category afterward (Elboher et al., 2020). Similarly, networks can be reduced using approximative bisimulations (Prabhakar, 2022). These works are extended using interval neural networks (Prabhakar & Afzal, 2019; Sotoudeh & Thakur, 2020; Boudardara et al., 2022; 2023a;b), where the weights of a neural network are replaced with intervals during the sound reduction. It is worth mentioning that the reduced network can be re-enlarged using residual reasoning (Elboher et al., 2022). For ReLU networks, it is also possible to merge neurons that are entirely in the nonpositive or nonnegative region, respectively, without inducing outer approximations (Zhong et al., 2023). Network reductions can also be achieved by clustering similar neurons for inputs of a given dataset (Ashok et al., 2020); however, $80-90\%$ of the neurons remain when formal error bounds are demanded. Convolutional neural networks with max-pooling layers can be reduced by pruning branches leading up to pooling neuron and providing respective error bounds (Ostrovsky et al., 2022). Most approaches only consider relatively small networks with ReLU neurons, while some approaches also consider sound network reduction for specific other activation functions such as tanh (Boudardara et al., 2023c).

## 1.2 Contributions

To summarize, our main contributions are as follows:

- We present a novel, fully automatic approach to soundly reduce large neural networks by merging similar neurons for a given specification.

- The reduced network is constructed on the fly, and the verification of the reduced network entails the verification of the original network.

- As opposed to related work, our approach is applicable to all neural networks with element-wise activation functions, including ReLU, sigmoid, and tanh.

- The networks considered in the evaluation have up to an order of magnitude more neurons in a single layer than previous sound network reduction approaches considered in total, demonstrating the scalability of our approach.

We demonstrate our approach using reachability analysis with zonotopes (Girard, 2005; Singh et al., 2018a). The extension to other set-based verification tools is straightforward. The resulting reduced network can also be exported and verified using optimization-based verification tools (Sec. 1.1).

The remainder of this work is structured as follows: Sec. 2 introduces the notation and background for this work. Then, we present our novel, fully automatic, sound network reduction approach in Test. 3. In Test. 4, we discuss applications of our approach, including the reduction of large convolutional neural networks and how the reduced network can be reused for changed specifications. Finally, we evaluate our approach in Test. 5 and conclude in Test. 6.

## 2 Preliminaries

### 2.1 Notation

We denote scalars and vectors by lower-case letters, matrices by upper-case letters, and sets by calligraphic letters. The $i$-th element of a vector $v \in \mathbb{R}^n$ is written as $v_{(i)}$, and the element in the $i$-th row and $j$-th column of a matrix $A \in \mathbb{R}^{n \times m}$ is written as $A_{(i,j)}$. The $i$-th row and $j$-th column are written as $A_{(i,\cdot)}$ and $A_{(\cdot,j)}$, respectively. The concatenation of two matrices $A$ and $B$ is denoted by $[A\ B]$. We write $\mathbb{R}_+$ to refer to all positive real numbers. For $n \in \mathbb{N}$, we denote the identity matrix by $I_n$, and we use the notation $[n] = \{1, \ldots, n\}$. Let $\mathcal{C} \subseteq [n]$, then $A_{(\mathcal{C},\cdot)}$ extracts all rows $i \in \mathcal{C}$ in lexicographic order. We denote the cardinality of a discrete set $\mathcal{C}$ by $|\mathcal{C}|$. Let $\mathcal{S} \subset \mathbb{R}^n$ be a continuous set, then $\mathcal{S}_{(i)}$ is its projection on the $i$-th dimension. The symbols $\mathbf{0}$ and $\mathbf{1}$ refer to matrices with all zeros and ones of proper dimensions, respectively. The set-based evaluation of a function $f \colon \mathbb{R}^n \to \mathbb{R}^m$ is written as $f(\mathcal{S}) = \{f(x) \mid x \in \mathcal{S}\}$. Given two sets $\mathcal{S}_1, \mathcal{S}_2$, then the Minkowski sum is denoted by $\mathcal{S}_1 \oplus \mathcal{S}_2 = \{s_1 + s_2 \mid s_1 \in \mathcal{S}_1,\ s_2 \in \mathcal{S}_2\}$. If either summand of the Minkowski sum is given as a vector, it is implicitly converted to a singleton. The Cartesian product is written as $\mathcal{S}_1 \times \mathcal{S}_2 = \{[s_1^T\ \ s_2^T]^T \mid s_1 \in \mathcal{S}_1, s_2 \in \mathcal{S}_2\}$. An interval with bounds $l, u \in \mathbb{R}^n$, where $l \leq u$ holds element-wise, is denoted by $[l, u]$. Unless otherwise stated, all continuous sets in this paper are represented as zonotopes except for the exact sets denoted by $\square^*$ and intervals denoted by $\mathcal{I}_\square$.

### 2.2 Neural Networks

We introduce feed-forward neural networks (Bishop & Nasrabadi, 2006, Sec. 5.1) in their standard form here and discuss the sound reduction of convolutional neural networks in Sec. 4.1.

**Definition 1** (Feed-Forward Neural Networks (Bishop & Nasrabadi, 2006, Sec. 5.1)). *Given $\kappa$ alternating linear and nonlinear layers and let $x \in \mathbb{R}^{n_0}$ be the input and $y \in \mathbb{R}^{n_\kappa}$ be the output of a neural network, we can formulate a neural network $\Phi \colon \mathbb{R}^{n_0} \to \mathbb{R}^{n_\kappa}$ with $y = \Phi(x)$ as follows:*

$$h_0 = x, \quad h_k = L_k(h_{k-1}), \quad y = h_\kappa, \qquad k \in [\kappa],$$

*where $L_k \colon \mathbb{R}^{n_{k-1}} \to \mathbb{R}^{n_k}$ computes the operation of layer $k$. Let $W_k \in \mathbb{R}^{n_k \times n_{k-1}}$, $b_k \in \mathbb{R}^{n_k}$, and $\phi_k(\cdot)$ be the respective continuous activation function (e.g., sigmoid and ReLU), which is applied element-wise, then*

$$L_k(h_{k-1}) = \begin{cases} W_k h_{k-1} + b_k & \text{if layer } k \text{ is linear,} \\ \phi_k(h_{k-1}) & \text{otherwise.} \end{cases}$$

Please note that some works formulate the layers of a neural network as $L_k(h_{k-1}) = \phi_k(W_k h_{k-1} + b_k)$, i.e., having the linear part and the nonlinear part within one layer. We opted to separate them into individual layers as the enclosure is handled differently (Prop. 3), however, the architecture is the same. Thus, we always have linear and nonlinear layers in alternating fashion.

The last linear and last nonlinear layers are called output layers, all other layers are called hidden layers. If all hidden layers output the same number of neurons, we write $6 \times 200$ to refer to a network with 6 linear and 6 nonlinear hidden layers with 200 neurons each.

### 2.3 Set-Based Computing

We use reachability analysis to verify neural networks: Let $\mathcal{X} \subset \mathbb{R}^{n_0}$ be the input set of a neural network $\Phi$. Then, the exact output set $\mathcal{Y}^* = \Phi(\mathcal{X})$ is computed by

$$\mathcal{H}_0^* = \mathcal{X}, \quad \mathcal{H}_k^* = L_k(\mathcal{H}_{k-1}^*), \quad \mathcal{Y}^* = \mathcal{H}_\kappa^*, \qquad k \in [\kappa]. \tag{1}$$

Usually, $\mathcal{X}$ is constructed by perturbing each dimension of an input $x \in \mathbb{R}^{n_0}$ up to a maximal perturbation radius $\epsilon \in \mathbb{R}_+$. Unfortunately, it is computationally infeasible to obtain these exact sets in general as the problem is NP-hard for networks with ReLU activations (Katz et al., 2017). For that reason, verification tools usually compute an outer approximation $\mathcal{Y} \supseteq \mathcal{Y}^*$ using a specific set representation. Our reduction approach works for any set representation that can be enclosed by an interval and that is closed under the Minkowski addition of intervals. We use zonotopes as an example to demonstrate our approach:

**Definition 2** (Zonotope (Girard, 2005, Def. 1))**.** *Given a center vector $c \in \mathbb{R}^n$ and a generator matrix $G \in \mathbb{R}^{n \times q}$, a zonotope is defined as*

$$\mathcal{Z} = \langle c, G \rangle_Z = \left\{ c + \sum_{j=1}^{q} \beta_j G_{(\cdot, j)} \;\middle|\; \beta_j \in [-1, 1] \right\}.$$

For zonotopes, the input set is given by $\mathcal{X} = \langle x, \epsilon I_{n_0} \rangle_Z$ and the required operations are computed as follows:

**Proposition 1** (Interval Enclosure (Althoff, 2010, Prop. 2.2))**.** *Given a zonotope $\mathcal{Z} = \langle c, G \rangle_Z$, the enclosing interval $[l, u] = \texttt{interval}(\mathcal{Z}) \supseteq \mathcal{Z}$ is given by*

$$l = c - \Delta g, \quad u = c + \Delta g, \qquad with \; \Delta g = \sum_{j=1}^{q} |G_{(\cdot, j)}|.$$

**Proposition 2** (Interval Addition (Althoff, 2010, Eq. 2.1))**.** *Given a zonotope $\mathcal{Z} = \langle c, G \rangle_Z \subset \mathbb{R}^n$ and an interval $\mathcal{I} = [l, u] \subset \mathbb{R}^n$,*

$$\mathcal{Z} \oplus \mathcal{I} = \langle c + c_{\mathcal{I}}, [G \; \text{Diag}(u - c_{\mathcal{I}})] \rangle_Z,$$

*where $c_{\mathcal{I}} = \frac{l+u}{2}$ and $\text{Diag}(\cdot)$ returns a diagonal matrix.*

### 2.4 Neural Network Verification

As we use zonotopes as an example to verify neural networks, we briefly introduce the main steps to propagate a zonotope through a neural network. These might differ if another verification engine is used. Since the propagation in (1) cannot be computed exactly in general, we enclose the output of each layer:

**Proposition 3** (Image Enclosure (Singh et al., 2018a, Sec. 3))**.** *Let $\mathcal{H}_{k-1} \supseteq \mathcal{H}_{k-1}^*$ be an input set to layer $k$, then*

$$\mathcal{H}_k = \texttt{enclose}(L_k, \mathcal{H}_{k-1}) \supseteq \mathcal{H}_k^*$$

*computes an outer-approximative output set.*

While zonotopes can be propagated through linear layers exactly (Girard, 2005), the propagation through nonlinear layers has to be outer-approximative to ensure soundness. The main steps to enclose the output of

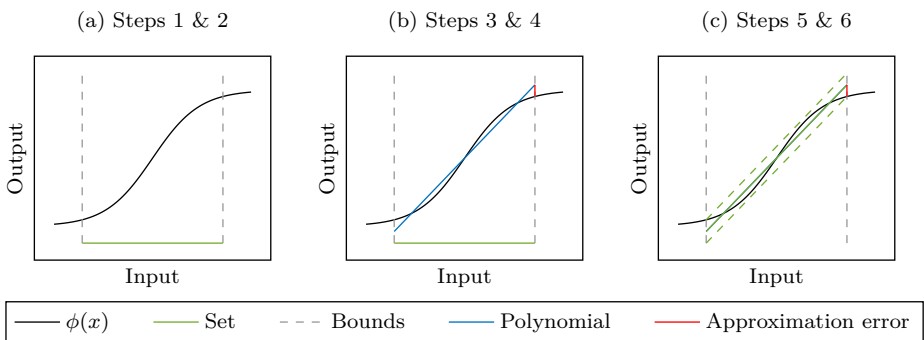

Figure 1: Main steps of enclosing a nonlinear layer. Step 1: Neuron-wise sigmoid function. Step 2: Find input bounds. Step 3: Approximate with linear function. Step 4: Determine approximation error. Step 5: Apply linear transformation on input. Step 6: Enclose using approximation error.

nonlinear layers are illustrated in Fig. 1: For each nonlinear layer, we iterate over all neurons $i$ in the current layer by projecting the input set $\mathcal{H}_{k-1}$ onto its $i$-th dimension (Step 1) and determining the input bounds using Prop. 1 (Step 2). We then find an approximating linear function within the input bounds (Bishop & Nasrabadi, 2006, Sec. 3) (Step 3). Crucially, soundness is ensured by bounding the approximation error (Step 4). Finally, we apply the linear transformation on $\mathcal{H}_{k-1(i)}$ to approximate the nonlinear layer (Step 5) and enclose the activation function using the approximation error (Step 6; Prop. 2). Thus, by propagating a given input set $\mathcal{X}$ through all layers of a neural network and enclosing their output sets using Prop. 3, we can enclose the exact output set of the entire network by $\mathcal{Y} = \mathcal{H}_\kappa \supseteq \mathcal{Y}^* = \Phi(\mathcal{X})$.

## 2.5 Problem Statement

Given an input set $\mathcal{X} \subset \mathbb{R}^{n_0}$, a neural network $\Phi$, and an unsafe set $\mathcal{S} \subset \mathbb{R}^{n_\kappa}$, we want to automatically construct a sound reduced network $\widehat{\Phi}$, for which the verification entails the verification of the original network for the given $\mathcal{X}$ and $\mathcal{S}$:

$$\widehat{\Phi}(\mathcal{X}) \cap \mathcal{S} = \emptyset \implies \Phi(\mathcal{X}) \cap \mathcal{S} = \emptyset.$$

This formulation covers all specifications from the VNN competition (Bak et al., 2021; Brix et al., 2023), and more complex specifications can also be formulated as reachability problems (Lopez et al., 2023a; Wright & Stark, 2020). Further, we want the construction and verification of $\widehat{\Phi}$ to be faster than verifying $\Phi$ directly.

## 3 Fully Automatic and Sound Neural Network Reduction

Our sound neural network reduction is based on the observation that many neurons in a layer $k$ behave similarly for a specific input $x \in \mathbb{R}^{n_0}$, e.g., many sigmoid neurons are fully saturated and thus output a value near 0 or 1 as shown in Fig. 2. Neuron saturation (Rakitianskaia & Engelbrecht, 2015) and neural activation patterns (Bäuerle et al., 2022) have been observed in the literature, however, to the best of our knowledge, they have not been exploited for the verification of neural networks. Please note that our approach is not restricted to the saturation values of an activation function.

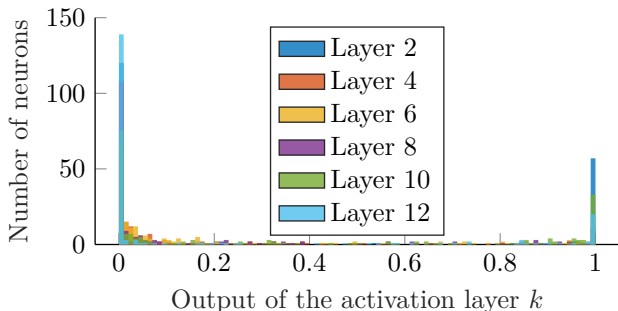

Figure 2: Sigmoid activations of a $6 \times 200$ neural network with an image input from the MNIST digit dataset. For a specific input $x$, many neurons output values close to the saturation values 0 and 1.

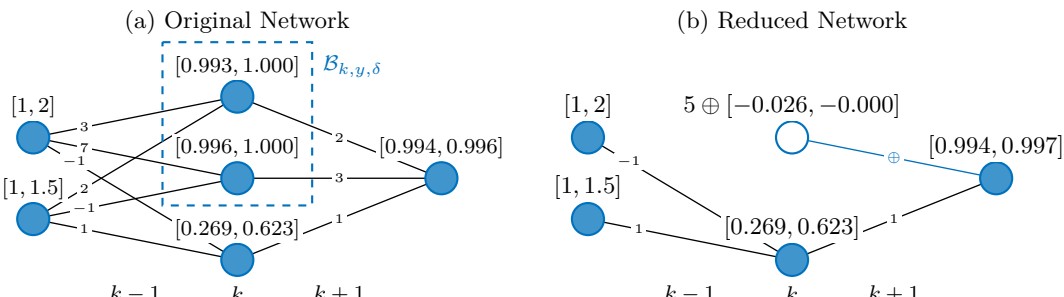

Figure 3: Reduction example on a sigmoid neural network using a single merge bucket $\mathcal{B}_{k,y,\delta}$ with $y = 1$ and $\delta = 0.01$: The shown bounds of each neuron cover the uncertainty after the sigmoid activation function is applied. All neurons within $\mathcal{B}_{k,y,\delta}$ are replaced by a new neuron with constant output $2y + 3y = 5$. Then, we bound the approximation error to obtain an outer approximation using Prop. 4.

## 3.1 Neuron Merging

Subsequently, we explain how similar neurons can
help to construct a reduced network $\widehat{\Phi}$, where the verification of $\widehat{\Phi}$ entails the verification of the original network $\Phi$. We gather the neurons with similar values using merge buckets:

**Definition 3** (Merge Buckets). *Given output bounds $\mathcal{I}_k \supseteq \mathcal{H}_k^*$ of a nonlinear layer $k \in [\kappa]$ with $n_k$ neurons, an output $y \in \mathbb{R}$, and a tolerance $\delta \in \mathbb{R}_+$, then a merge bucket is defined as*

$$\mathcal{B}_{k,y,\delta} = \left\{ i \in [n_k] \mid \mathcal{I}_{k(i)} \subseteq [y - \delta, y + \delta] \right\}.$$

Conceptually, we replace all neurons in a merge bucket $\mathcal{B}_{k,y,\delta}$ by a single neuron with constant output $y$ and adjust the weight matrices of the linear layers $k - 1$ and $k + 1$ such that the reduced network $\widehat{\Phi}$ approximates the behavior of the original network $\Phi$. Finally, we add an approximation error $\mathcal{I}'_{k+1} \subset \mathbb{R}^{n_{k+1}}$ to the output of the linear layer $k + 1$ to obtain a sound outer approximation (Fig. 3). To simplify the notation, we adapt Def. 1 for linear layers slightly to also include the approximation error:

$$\widehat{L}_k(h_{k-1}) = \begin{cases} W_k h_{k-1} + b_k \oplus \mathcal{I}'_k & \text{if layer } k \text{ is linear,} \\ \phi_k(h_{k-1}) & \text{otherwise,} \end{cases} \tag{2}$$

where $\mathcal{I}'_k$ is initialized with $[\mathbf{0}, \mathbf{0}] \subset \mathbb{R}^{n_k}$. With that, we can formally introduce how we merge neurons within the neural network:

**Proposition 4** (Neuron Merging). *Given a nonlinear hidden layer $k \in [\kappa]$ of a network $\Phi$, output bounds $\mathcal{I}_k \supseteq \mathcal{H}_k^* \subset \mathbb{R}^{n_k}$, a merge bucket $\mathcal{B} \subseteq [n_k]$, and the indices of the remaining neurons $\overline{\mathcal{B}} = [n_k]\backslash\mathcal{B}$, we construct a sound reduced network $\widehat{\Phi}$, where we adjust the nonlinear layer $k$ to only process the remaining neurons, and in the adjacent linear layers, we remove the merged neurons and bound the approximation error:*

$$\text{Layer } k - 1: \qquad \widehat{W}_{k-1} = W_{k-1(\overline{\mathcal{B}},\cdot)}, \qquad \widehat{b}_{k-1} = b_{k-1(\overline{\mathcal{B}})}, \qquad \widehat{\mathcal{I}}'_{k-1} = \mathcal{I}'_{k-1(\overline{\mathcal{B}})},$$

$$\text{and layer } k + 1: \qquad \widehat{W}_{k+1} = W_{k+1(\cdot,\overline{\mathcal{B}})}, \qquad \widehat{b}_{k+1} = b_{k+1}, \qquad \widehat{\mathcal{I}}'_{k+1} = W_{k+1(\cdot,\mathcal{B})}\mathcal{I}_{k(\mathcal{B})}.$$

*Proof.* The proof can be found in Appendix A. □

Fig. 3 illustrates the neuron merging for a toy neural network with sigmoid activation. Please note that we display the bounds for each neuron as intervals; however, the underlying verification engine might use a different set representation, e.g., zonotopes.

By iteratively applying Prop. 4, our approach can be naturally extended to multiple disjoint merge buckets:

$$\boldsymbol{\mathcal{B}}_{k,\delta} = \{\mathcal{B}_{k,y_1,\delta}, \mathcal{B}_{k,y_2,\delta}, \ldots\}. \tag{3}$$

The merging with multiple disjoint merge buckets can be done in parallel as the required adaptations of the adjacent linear layers do not interfere with each other. The overall approximation error is then given by the Minkowski sum of the individual approximation errors (Prop. 4):

$$\widehat{\mathcal{I}}'_{k+1} \overset{(3)}{=} \bigoplus_{\mathcal{B}\in\boldsymbol{\mathcal{B}}} \widehat{\mathcal{I}}'_{k+1,\mathcal{B}} \overset{(\text{Prop. 4})}{=} \bigoplus_{\mathcal{B}\in\boldsymbol{\mathcal{B}}} W_{k+1(\cdot,\mathcal{B})}\mathcal{I}_{k(\mathcal{B})} \overset{(\text{disjoint }\boldsymbol{\mathcal{B}})}{=} W_{k+1(\cdot,\bigcup_{\mathcal{B}\in\boldsymbol{\mathcal{B}}}\mathcal{B})}\mathcal{I}_{k(\bigcup_{\mathcal{B}\in\boldsymbol{\mathcal{B}}}\mathcal{B})}. \tag{4}$$

### 3.2 Initialization of Merge Buckets

We define two different methods to initialize merge buckets:

**Static buckets** The merge buckets are determined by the asymptotic values of the respective activation function $\phi_k$ of a nonlinear layer $k$:

$$\boldsymbol{\mathcal{B}}_{k,\delta} = \begin{cases} \{\mathcal{B}_{k,0,\delta},\ \mathcal{B}_{k,1,\delta}\} & \text{if } \phi_k = \text{sigmoid}, \\ \{\mathcal{B}_{k,-1,\delta},\ \mathcal{B}_{k,1,\delta}\} & \text{if } \phi_k = \tanh, \\ \{\mathcal{B}_{k,0,\delta}\} & \text{if } \phi_k = \text{ReLU}. \end{cases} \tag{5}$$

For ReLU layers, setting $\delta = 0$ and using static merge buckets results in no approximation error similar to the approach in (Zhong et al., 2023), as only neurons with entirely negative input for the given input set $\mathcal{X}$ are removed.

**Dynamic buckets** The merge buckets are dynamically initialized using the center of the bounds $\mathcal{I}_k = [l_k, u_k] \subset \mathbb{R}^{n_k}$ of each neuron:

$$\boldsymbol{\mathcal{B}}_{k,\delta} = \left\{ \mathcal{B}_{k,c_{(i)},\delta} \ \middle|\ c = \tfrac{l_k+u_k}{2},\ i \in [n_k] \right\}, \tag{6}$$

where we ensure that the buckets are disjoint and are only used if they contain multiple neurons. Please note that the buckets could also be created using clustering algorithms (Ashok et al., 2020); however, we choose the center of each neuron directly to obtain a linear computational overhead. The computational overhead of clustering algorithms might be negligible for other underlying verification engines than the zonotope approach considered in this work.

### 3.3 Automatic Determination of Bucket Tolerances

The bucket tolerance $\delta \in \mathbb{R}_+$ influences how many neurons are merged, where a larger value results in more aggressive neuron merging and thus a larger outer approximation. However, determining a good value for $\delta$ is tedious as it is not immediately clear how much the network is reduced for any given value for $\delta$. Thus, we automatically determine $\delta$ given a desired remaining number of neurons in Alg. 1 using binary search. We denote the ratio of remaining neurons compared to the original network with the reduction rate $\rho \in [0,1]$. To verify a given specification, we initially choose a very small $\rho$ and iteratively increase it if the reduction is too outer-approximative. This realizes us to verify many specifications using a heavily reduced network (Test. 5), and thus to a similar degree, the verification time is reduced. Once $\rho = 1$ is reached, the original network is used and no reduction is applied.

---

**Algorithm 1** Automatic Determination of Bucket Tolerance

---
**Require:** Bounds $\mathcal{I}_k = [l_k, u_k]$, reduction rate $\rho$

1:   $\delta_{\min} \leftarrow 0, \delta_{\max} \leftarrow \max(u_k) - \min(l_k)$

2:   **do**                $\triangleright$ 1) Find upper bound for bucket tolerance

3:       $\delta_{\max} \leftarrow 10 * \delta_{\max}$

4:       Initialize merge buckets $\mathcal{B}_{k,\delta_{\max}}$          $\triangleright$ Sec. 3.2

5:       $\widehat{n}_k \leftarrow n_k - \sum_{\mathcal{B} \in \mathcal{B}_{k,\delta_{\max}}} |\mathcal{B}|$         $\triangleright$ Remaining neurons

6:   **while** $\widehat{n}_k/n_k > \rho$

7:   **do**                                 $\triangleright$ 2) Binary search

8:       $\delta \leftarrow (\delta_{\min} + \delta_{\max})/2$

9:       Initialize merge buckets $\mathcal{B}_{k,\delta}$            $\triangleright$ Sec. 3.2

10:      $\widehat{n}_k \leftarrow n_k - \sum_{\mathcal{B} \in \mathcal{B}_{k,\delta}} |\mathcal{B}|$           $\triangleright$ Remaining neurons

11:      **if** $\widehat{n}_k < \rho n_k$ **then**

12:         $\delta_{\max} \leftarrow \delta$                $\triangleright$ Too many neurons merged

13:      **else**

14:         $\delta_{\min} \leftarrow \delta$                 $\triangleright$ Too few neurons merged

15:      **end if**

16:   **while** $\widehat{n}_k/n_k \not\approx \rho$

17:   **return** $\mathcal{B}_{k,\delta}$

---

## 3.4   On-the-fly Neural Network Reduction

Please note that we require output bounds $\mathcal{I}_k$ of the next nonlinear layer $k$ to merge neurons with similar values using Prop. 4. However, computing them requires the construction of high-dimensional zonotopes via the linear layer $k - 1$ and the propagation of the zonotopes through the nonlinear layer $k$, where we have to compute the image enclosure for all neurons (Prop. 3) – which is what should be avoided. Thus, we deploy a one-step look-ahead algorithm (Alg. 2) using interval arithmetic (Jaulin et al., 2001) to avoid these expensive computations and reduce the network on the fly. As the look-ahead is just a single step, the computed bounds are tight and do not contribute to the wrapping effect.

---

**Algorithm 2** On-the-fly Neural Network Reduction

---
**Require:** Input $\mathcal{X}$, neural network layers $L_k$, $k \in [\kappa]$, reduction rate $\rho$

1:   $\mathcal{H}_0 \leftarrow \mathcal{X},\ \widehat{L}_1 \leftarrow L_1$

2:   **for** $k = 2, 4, \ldots, \kappa$ **do**

3:      **if** $k < \kappa$ **then**                   $\triangleright$ 1) Look ahead

4:         $\mathcal{I}_{k-2} \leftarrow \mathtt{interval}\,(\mathcal{H}_{k-2})$           $\triangleright$ Prop. 1

5:         $\mathcal{I}_k \leftarrow L_k(\widehat{L}_{k-1}(\mathcal{I}_{k-2}))$

6:         Determine merge buckets $\mathcal{B}_{k,\delta}$       $\triangleright$ Sec. 3.3

7:         $\widehat{L}_{k-1}, \widehat{L}_k, \widehat{L}_{k+1} \leftarrow$ Merge         $\triangleright$ Prop. 4

8:      **end if**

9:                              $\triangleright$ 2) Verify reduced network

10:     $\mathcal{H}_{k-1} \leftarrow \mathtt{enclose}(\widehat{L}_{k-1}, \mathcal{H}_{k-2})$       $\triangleright$ Prop. 3

11:     $\mathcal{H}_k \leftarrow \mathtt{enclose}(\widehat{L}_k, \mathcal{H}_{k-1})$

12:   **end for**

13:   **return** $\mathcal{Y} \leftarrow \mathcal{H}_\kappa$

---

We summarize Alg. 2 subsequently: Instead of propagating the zonotope $\mathcal{H}_{k-2}$ itself forward, we just propagate interval bounds of $\mathcal{H}_{k-2}$ to the next nonlinear layer $k$ (Alg. 2-5). Although intervals are not closed under the linear map, the output bounds of the linear layer $k - 1$ are tight, and the propagation through the nonlinear layer $k$ does not induce additional outer approximations. This realizes a tight computation of the output bounds $\mathcal{I}_k$ with negligible computational overhead; a discussion on the overhead can be found in

Appendix A. After $\mathcal{I}_k$ is obtained, the merge buckets are determined (Alg. 2), and the network is reduced by merging the respective neurons (Alg. 2). Finally, we propagate the zonotope $\mathcal{H}_{k-2}$ through the reduced layers. Thus, we never construct a high-dimensional zonotope during the verification. Note that the number of input and output neurons remains unchanged.

**Theorem 1** (Sound Neural Network Reduction). *Given an input set $\mathcal{X}$, a neural network $\Phi$, and a reduction rate $\rho$, Alg. 2 constructs a reduced network $\widehat{\Phi}_\rho$ satisfying the problem statement in Sec. 2.5.*

*Proof.* The proof can be found in Appendix A. □

## 4 Applications

In this section, we discuss additional applications of our novel neural network reduction approach.

### 4.1 Reduction of Convolutional Neural Networks and Sound Image Compression

Convolutional neural networks are obtaining state-of-the-art results for image classification tasks (Li et al., 2022). However, neural networks for image classification are typically very large and thus particularly hard to verify. We show in this section that our novel neural network reduction approach can be naturally extended to convolutional networks. Let us start by introducing the main layer within a convolutional network:

**Definition 4** (Convolutional Layer (Bishop & Nasrabadi, 2006, Sec. 5.5.6)). *Given an input $I \in \mathbb{R}^{c_I \times h_I \times w_I}$ and a kernel $K \in \mathbb{R}^{c_O \times c_I \times h_K \times w_K}$, a convolutional layer computes the output $O \in \mathbb{R}^{c_O \times h_O \times w_O}$ for $k \in [c_O]$, $i \in [h_O]$, $j \in [w_O]$ as follows:*

$$O_{(k,i,j)} = \sum_{l=1}^{c_I} \sum_{m=1}^{h_K} \sum_{n=1}^{w_K} K_{(k,l,m,n)} I_{(l,i+m,j+n)},$$

*where $h_O = h_I - (h_K - 1)$, $w_O = w_I - (w_K - 1)$ and $c_I, c_O$ are the number of input and output channels, respectively.*

Convolutional layers can be viewed as linear layers as defined in Def. 1 with shared weights (Bishop & Nasrabadi, 2006, Sec. 5.5.6). The exact construction of such a linear layer is given in Appendix A. Thus, our approach is also directly applicable for convolutional layers. An analogous conversion can be done for other typical layers within a convolutional network, such as subsampling and average pooling layers (Bishop & Nasrabadi, 2006, Sec. 5.5.6).

One important property of convolutional networks is the preservation of neighborhood: As the same kernel is applied to the entire input, pixels of the output have similar values if the respective pixels in the input have similar values. Neighboring pixels having similar values are typical in the field of image classification because many images contain large areas or objects with a similar color. For example,

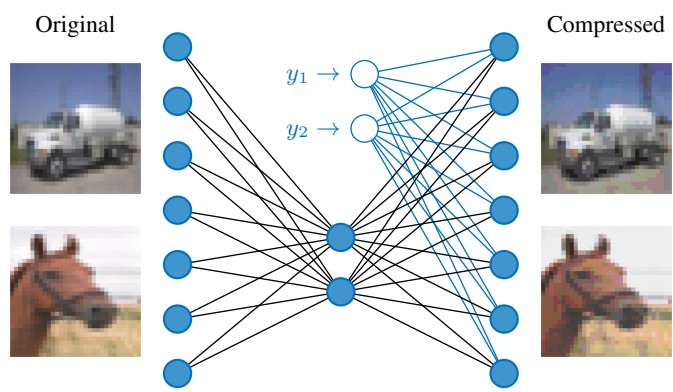

Figure 4: Visualization of CIFAR-10 images: Original images (left) and corresponding compressed images (right), where we set all neurons within the same merge bucket $\mathcal{B}_{k,y_i,\delta} \in \mathcal{B}$ to value $y_i$ to extract the compressed image without added approximation error. Verifying the compressed image with formal error bounds drastically reduces the number of neurons of hidden layers.

the sky has similar shades of blue, and traffic signs typically have only one background color and one foreground color. To the best of our knowledge, our approach is the first to explicitly exploit this property of

convolutional networks for sound neural network reduction. Intuitively, an uncertain image is compressed into superpixels with formal error bounds. Let us demonstrate this by an example: CIFAR-10 images require $32 \times 32 \times 3 = 3072$ input neurons to the network. However, many of these pixels have very similar values (Fig. 4). Thus, using our approach with dynamic merge buckets (Sec. 3.2), we can compress an image with formal error bounds as follows:

**Corollary 1** (Sound Compression). *Given an uncertain image $\mathcal{X} \subset \mathbb{R}_+^{c_I \times h_I \times w_I}$ and a reduction rate $\rho \in [0, 1]$, we can construct a neural network $\widehat{\Phi}_\rho$ that compresses this image with formal error bounds as follows: Let $K \in \mathbb{R}^{c_I \times c_I \times 1 \times 1}$ be a kernel of a convolutional layer, where*

$$K_{(k,l,1,1)} = \begin{cases} 1 & \text{for } k = l, \\ 0 & \text{otherwise,} \end{cases} \qquad k, l \in [c_I], \tag{7}$$

*and $\Phi$ be a neural network with two convolutional layers with kernel $K$ and one ReLU activation. The reduced network $\widehat{\Phi}_\rho$, obtained by applying Thm. 1 using $\mathcal{X}$, $\rho$, and dynamic merge buckets compresses the input $\mathcal{X}$ with sound error bounds according to $\rho$.*

*Proof.* The proof can be found in Appendix A. $\square$

In the truck example in Fig. 4, for a perturbation radius $\epsilon = 0.01$ and a reduction rate $\rho = 0$, all 3072 neurons of the hidden layer of the compression network $\widehat{\Phi}_\rho$ are dynamically merged using 21 merge buckets. Thus, the image is compressed into a 21-dimensional space in the hidden layer of $\widehat{\Phi}_\rho$ and then re-enlarged to 3072 neurons in the output layer with added approximation error. Please note that usually, the image is not re-enlarged and the compressed image is passed to the actual network (see below), where we again merge similar neurons using our approach – we just do this here for illustration purposes (Fig. 4): Due to the three color channels, the original truck image has 983 unique colors, which get compressed into 178 unique colors with formal error bounds. Similarly, the original horse image has 891 unique colors, which get compressed into 142 unique colors, again using 21 dynamic merge buckets.

In larger convolutional networks, a similar compression happens in each hidden layer; however, these are usually not as easy to grasp visually due to the increased number of channels in hidden layers. Note that we can prepend the layers of the compression network defined in Cor. 1 to any network as a preprocessing step to reduce the input dimension. The required steps for this preprocessing are provided in Alg. 3: We first construct the compression network as in Cor. 1. As we only compute the layers of the reduced network in Alg. 3, we only require the input set represented as an interval $\mathcal{I}_\mathcal{X}$, which is usually the case for many benchmarks (Brix et al., 2024). Thus, we can construct a new low-dimensional input $\mathcal{H}_{-2}$ represented by the used set representation according to the remaining neurons. Thus, we are not required to initialize the high-dimensional input set using a more complex set representation, which usually speeds up the computation. In our case using zonotopes, this preprocessing step results in much fewer generators (Prop. 2) and thus speeds up the computation. The output set is computed in Alg. 3 by propagating the set through all remaining layers and reducing them on the fly (Alg. 2). Due to the on-the-fly reduction, the more complex set representation is kept in a low-dimensional space as by the time it arrives at a given layer, this layer is already reduced (Thm. 1). This becomes increasingly beneficial with the complexity of the set representation used to verify the network. Note that Alg. 3 works on any neural network, but is especially beneficial for convolutional networks because neighboring pixels often have similar values.

---

**Algorithm 3** Sound Compression Preprocessing

---
**Require:** Input $\mathcal{I}_\mathcal{X}$, neural network $\Phi_{\text{org}}$, reduction rate $\rho$
1: $L_{-2}, L_{-1}, L_0 \leftarrow$ Construct $\Phi_{\text{pre}}$                  $\triangleright$ Cor. 1
2: $\widehat{L}_{-2}, \widehat{L}_{-1}, \widehat{L}_0 \leftarrow$ Reduce using $\mathcal{I}_\mathcal{X}, \rho$              $\triangleright$ Thm. 1
3: $\mathcal{H}_{-2} \leftarrow \texttt{zonotope}(\widehat{L}_{-2}(\mathcal{I}_\mathcal{X}))$              $\triangleright$ Prop. 2
4: $\Phi'_{\text{org}}(\cdot) \leftarrow \Phi_{\text{org}}(\widehat{L}_0(\widehat{L}_{-1}(\cdot)))$            $\triangleright$ Prepend layers
5: $\mathcal{Y} \leftarrow$ Execute Alg. 2 using $\mathcal{H}_{-2}, \Phi'_{\text{org}}, \rho$           $\triangleright$ Thm. 1
6: **return** $\mathcal{Y}$

---

## 4.2 Reusing Reduced Networks

In general, our approach requires the computation of a new reduced neural network for different input sets. In this section, we highlight several applications where the reduced networks can be reused nevertheless:

### 4.2.1 Branch-and-bound

Current state-of-the-art tools, e.g., all top-ranked tools of the last VNN competition (Brix et al., 2023), verify a neural network by applying various branch-and-bound algorithms in the verification. Branch-and-bound algorithms (Bunel et al., 2020) partition the verification problem into multiple simpler subproblems, solve them individually, and aggregate the results to reason about the overall problem (Sec. 1.1). Our novel reduction approach is orthogonal to these branch-and-bound algorithms and thus can be combined with them. Since the reduced network does not depend on using a specific set representation, we can reuse the reduced network on all subsets of the input set:

**Corollary 2** (Reusing Reduced Network on Subsets). *Given a neural network $\Phi$, an input set $\mathcal{X}$, and a reduction rate $\rho$, then a reduced network $\widehat{\Phi}_\rho$ according to Thm. 1 can be reused for all $\mathcal{X}' \subseteq \mathcal{X}$.*

*Proof.* The proof can be found in Appendix A. □

### 4.2.2 Export of reduced network

As the reduced network can be reused as described above, we provide an interface to export a reduced network for later usage, e.g., to verify the reduced network using another verification tool. Please note that a reduced network is of the form given in (2), having the approximation error added to the output of linear layers. As these approximation errors can be seen as additional inputs to the network, most verifiers can verify networks of this form, including optimization-based verifiers. In contrast, related work using interval neural networks (Prabhakar & Afzal, 2019; Sotoudeh & Thakur, 2020; Boudardara et al., 2022; 2023a;b) have intervals on the weight matrices of the network, which is much harder to integrate into existing verifiers.

### 4.2.3 Closed-loop verification

In closed-loop scenarios, a neural network is used as a controller in a dynamic system, which is updated every $\Delta t$. While branch-and-bound strategies work well in open-loop verification, other techniques are more common in closed-loop scenarios (Lopez et al., 2024). The issue with branch-and-bound strategies is that each subset has to be propagated according to a differential equation until the next network evaluation (every $\Delta t$), where each subset might again get splitted. Therefore, many techniques use more sophisticated set representation (Kochdumper et al., 2023; Bogomolov et al., 2019; Lopez et al., 2023b) and improve the abstraction by enclosing nonlinear functions with higher-order polynomials (Ladner & Althoff, 2023).

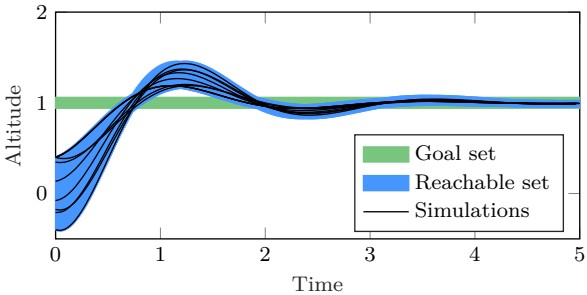

Figure 5: Quadrotor example (Lopez et al., 2024, Fig. 15): As the reachable set stays within a given goal set, a reduced network can be reused as long as the reachable set stays within the goal set.

One frequent goal in closed-loop verification is to show the stability of a given dynamic system over a specified time horizon. For example, the QUAD benchmark in the last ARCH competition (Lopez et al., 2023a) requires showing the stability of a neural-network-controlled quadrotor at a given altitude (Fig. 5). We can infer from the simulations that the state of the system barely changes over the last second. Thus, we can slightly enlarge the current reachable set at $t = 4s$ and use it to reduce the size of the network. This reduced network can then be reused in subsequent evaluations if the reachable set stays within the set used to reduce the network controller (Cor. 2).

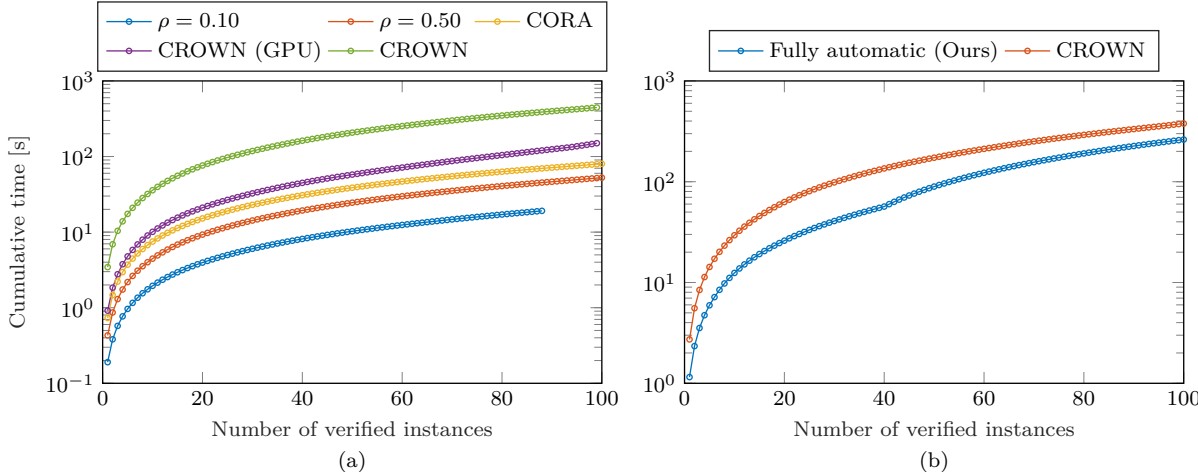

Figure 6: Comparison of our approach with varying reduction rates $\rho$ to CORA and $\alpha, \beta$-CROWN: (a) ERAN $6 \times 200$ sigmoid network on MNIST with varying reduction rates, (b) Marabou large convolutional network on CIFAR-10 using our fully automatic verification approach. Be aware of the log scale.

## 5 Evaluation

We implemented our approach in the MATLAB toolbox CORA (Althoff, 2015; Singh et al., 2018a) and evaluate it using several benchmarks and neural network variants from the VNN competition (Bak et al., 2021). The main results are presented in this section. All evaluation details, additional experiments, and ablation studies are given in Appendix B.

**Main results.** Our goal is to verify given queries very quickly to make formal verification tools applicable in time-critical scenarios, e.g., video streaming, where fast verification results are demanded. However, even for relatively small networks, the verification times for non-trivial queries can be comparatively long. For example, on the ERAN $6 \times 200$ sigmoid network classifying MNIST images, verification takes on average 1.5s per instance using the state-of-the-art tool $\alpha, \beta$-CROWN (Xu et al., 2020; Zhang et al., 2018; Wang et al., 2021) to obtain initial bounds using a GPU, which might not always be available in embedded systems. Without a GPU, this can take significantly longer (4.5s in this case). With our novel network reduction, we can verify most instances with only $\rho = 10\%$ of the neurons (Fig. 6a), such that the overall verification time – including reducing the network – is reduced to just 200ms on a CPU ($-96\%$ verification time).

For instances that cannot be verified with so few neurons, the reduction rate $\rho$ is iteratively increased until the instance is verified or the original network ($\rho = 1$) is reached. We show this iterative process on the CIFAR-10 dataset in Fig. 6b using the large convolutional neural network taken from the Marabou benchmark. Initially, most instances can be verified very quickly using a small reduction rate $\rho$. For these instances, the verification time is substantially faster than $\alpha, \beta$-CROWN ($-60\%$ verification time). If an instance cannot be verified at this abstraction level, $\rho$ is iteratively increased, which results in slightly longer verification times. Such a jump in verification time is clearly visible after $\sim 40$ instances. Overall, our fully automatic reduction approach is still 30% faster (Fig. 6). Please visit Sec. B.2 for an ablation study on this topic.

To obtain a better understanding on which neurons are merged throughout the verification process, we show in Fig. 7 the average remaining number of neurons of different network architectures while still verifying the images. The experiments show that large reduction rates are possible without sacrificing verifiability, especially for convolutional neural networks and including various activation functions.

This is in stark contrast to related work, where usually much smaller networks with only ReLU activation are analyzed. For example, $80 - 90\%$ of the neurons of the ERAN networks remain in related work once formal guarantees are demanded (Ashok et al., 2020, Fig. 2 & Tab. 2), whereas we reduce it to just $\sim 30\%$ for the ERAN networks (Fig. 7a) and $10 - 15\%$ for the Marabou networks (Fig. 7b). Additionally, our approach can

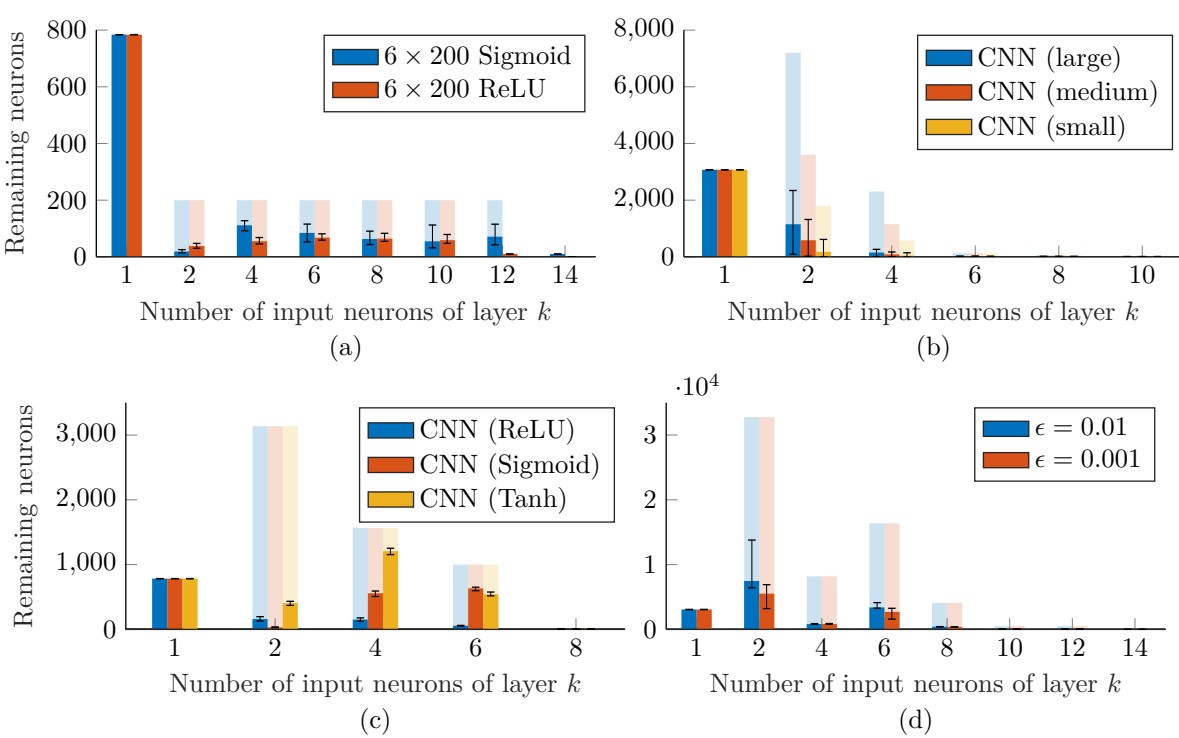

Figure 7: Average number of remaining neurons of networks taken from the (a) ERAN and (b) Marabou, (c) ERAN CNN variants, and (d) Cifar2020 benchmarks.

handle networks with any type of element-wise activation function (Fig. 7c)[1], and can handle much larger networks (Fig. 7d, shown with different perturbation radii $\epsilon$). Specifically, the network from the Cifar2020 benchmark has more neurons in a single layer than related work consider in total (Boudardara et al., 2023c).

**Additional experiments.** In Appendix B, we analyze each component of our approach in an ablation study and conduct additional experiments, including on sound image compression preprocessing, non-image inputs, and closed-loop verification.

## 6 Conclusion

We present a fully automatic and sound reduction approach to verify neural networks. Our approach considers the given specification during the reduction, which enables larger reductions than related work without compromising verifiability. All parameters of our approach are automatically tuned to minimize verification time, which greatly improves the usability of our approach. Additionally, while related work mostly consider networks with ReLU activations, our approach is agnostic towards the type of activation function as the reduction is realized using the respective output bounds. This simple yet effective approach is demonstrated in an extensive evaluation and ablation studies, where the verification time is reduced by up to 96%. These results are achieved on networks much larger than previously considered in sound neural network reduction. Moreover, we show how our reduced network can be reused despite its restriction on the input set during branch-and-bound algorithms and closed-loop verification. We hope this research advances the safe deployment of neural networks in safety- and time-critical environments.

---

[1] ERAN network variants taken from the ERAN website: `https://github.com/eth-sri/eran`

## Acknowledgements

The authors gratefully acknowledge partial financial support from the project FAI under project number 286525601 and the project SFB 1608 under project number 501798263, both funded by the German Research Foundation (Deutsche Forschungsgemeinschaft, DFG). We also want to thank Florian Finkeldei, Lukas Koller, and Mark Wetzlinger from our research group for their revisions of the manuscript.

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

# Appendix

## A   Proofs

In this section, we give the missing proof of the main body of the paper:

**Proposition 4** (Neuron Merging). *Given a nonlinear hidden layer $k \in [\kappa]$ of a network $\Phi$, output bounds $\mathcal{I}_k \supseteq \mathcal{H}_k^* \subset \mathbb{R}^{n_k}$, a merge bucket $\mathcal{B} \subseteq [n_k]$, and the indices of the remaining neurons $\overline{\mathcal{B}} = [n_k] \backslash \mathcal{B}$, we construct a sound reduced network $\widehat{\Phi}$, where we adjust the nonlinear layer $k$ to only process the remaining neurons, and in the adjacent linear layers, we remove the merged neurons and bound the approximation error:*

$$Layer\ k-1: \qquad \widehat{W}_{k-1} = W_{k-1(\overline{\mathcal{B}},\cdot)}, \qquad \widehat{b}_{k-1} = b_{k-1(\overline{\mathcal{B}})}, \qquad \widehat{\mathcal{I}}'_{k-1} = \mathcal{I}'_{k-1(\overline{\mathcal{B}})},$$

$$and\ layer\ k+1: \qquad \widehat{W}_{k+1} = W_{k+1(\cdot,\overline{\mathcal{B}})}, \qquad \widehat{b}_{k+1} = b_{k+1}, \qquad \widehat{\mathcal{I}}'_{k+1} = W_{k+1(\cdot,\mathcal{B})}\mathcal{I}_{k(\mathcal{B})}.$$

*Proof. Soundness.* We show that the output $\widehat{\mathcal{H}}_{k+1}$ of layer $k+1$ of the reduced network $\widehat{\Phi}$ is an outer approximation of the exact set $\mathcal{H}_{k+1}^*$:

$$\mathcal{H}_{k+1}^* \overset{(1)}{=} L_{k+1}\left(L_k\left(L_{k-1}(\mathcal{H}_{k-2}^*)\right)\right) \overset{(2)}{=} L_{k+1}\left(L_k\left(W_{k-1}(\mathcal{H}_{k-2}^*) + b_{k-1} \oplus \mathcal{I}'_{k-1}\right)\right).$$

Without loss of generality, we relabel the neurons such that $\mathcal{B} := [|\mathcal{B}|]$ and partition the neurons of layer $k$ using the Cartesian product:

$$\mathcal{H}_{k+1}^* = L_{k+1}\left(L_k\left((W_{k-1(\mathcal{B},\cdot)}\mathcal{H}_{k-2}^* + b_{k-1(\mathcal{B})} \oplus \mathcal{I}'_{k-1(\mathcal{B})}) \times (W_{k-1(\overline{\mathcal{B}},\cdot)}\mathcal{H}_{k-2}^* + b_{k-1(\overline{\mathcal{B}})} \oplus \mathcal{I}'_{k-1(\overline{\mathcal{B}})})\right)\right)$$

Rewriting the merged neurons (first term) using (1) and the remaining neurons (second term) using $\widehat{L}_{k-1}, \widehat{L}_k$ as defined in Prop. 4 results in:

$$\mathcal{H}_{k+1}^* = L_{k+1}\left(L_k\left(\mathcal{H}_{k-1(\mathcal{B})}^* \oplus \mathcal{I}'_{k-1(\mathcal{B})}\right) \times \widehat{L}_k\left(\widehat{L}_{k-1}(\mathcal{H}_{k-2}^*)\right)\right).$$

We then enclose all merged neurons by the given interval bounds:

$$\mathcal{H}_{k+1}^* \overset{(\text{Def. 3})}{\subseteq} L_{k+1}\left(\mathcal{I}_{k(\mathcal{B})} \times \widehat{L}_k\left(\widehat{L}_{k-1}(\mathcal{H}_{k-2}^*)\right)\right) =: \mathcal{H}'_{k+1},$$

and propagate them forward to the next nonlinear layer $k+1$. This operation implicitly propagates the new constant neuron forward to the bias of the layer $k+1$ as well without inducing additional outer approximations. Thus, using the identity $W(\tilde{\mathcal{I}}_1 \times \tilde{\mathcal{I}}_2) = W_{(\cdot,\mathcal{B})}\tilde{\mathcal{I}}_1 \oplus W_{(\cdot,\overline{\mathcal{B}})}\tilde{\mathcal{I}}_2$, we obtain:

$$\mathcal{H}'_{k+1} \overset{(\text{Def. 1})}{=} W_{k+1}\left(\mathcal{I}_{k(\mathcal{B})} \times \widehat{L}_k\left(\widehat{L}_{k-1}(\mathcal{H}_{k-2}^*)\right)\right) + b_{k+1}$$

$$= \left(W_{k+1(\cdot,\mathcal{B})}\mathcal{I}_{k(\mathcal{B})} \oplus W_{k+1(\cdot,\overline{\mathcal{B}})}\widehat{L}_k\left(\widehat{L}_{k-1}(\mathcal{H}_{k-2}^*)\right)\right) + b_{k+1}.$$

Finally, rearranging the terms obtains $\widehat{L}_{k+1}$ as defined in Prop. 4:

$$\mathcal{H}'_{k+1} = (W_{k+1(\cdot,\overline{\mathcal{B}})}\widehat{L}_k\left(\widehat{L}_{k-1}(\mathcal{H}_{k-2}^*)\right) + b_{k+1} \oplus W_{k+1(\cdot,\mathcal{B})}\mathcal{I}_{k(\mathcal{B})}$$

$$= \widehat{L}_{k+1}\left(\widehat{L}_k\left(\widehat{L}_{k-1}(\mathcal{H}_{k-2}^*)\right)\right),$$

which we can enclose using the underlying verification engine:

$$\mathcal{H}'_{k+1} \overset{(\text{Prop. 3})}{\subseteq} \widehat{\mathcal{H}}_{k+1}.$$

Hence, $\mathcal{H}_{k+1}^* \subseteq \mathcal{H}'_{k+1} \subseteq \widehat{\mathcal{H}}_{k+1}$. $\qquad\qquad\square$

**Overhead complexity of Alg. 2.** To analyze the complexity of the overhead due to bound propagation in Alg. 2 (Alg. 2 to 2), let us introduce some notations: Given two functions $f, g$, we use the notation $f(x) \in \mathcal{O}(g(x)) \iff \limsup_{x \to \infty} \frac{|f(x)|}{|g(x)|} < \infty$, and $f(x) \in \Omega(g(x)) \iff \limsup_{x \to \infty} \frac{|f(x)|}{|g(x)|} > 0$. Then, the computation of bounds using intervals through both layers (Alg. 2) can be done in $\mathcal{O}(n_{k-1} n_{k-2})$. In contrast, propagating the zonotope $\mathcal{H}_{k-2} \subset \mathbb{R}^{n_{k-2}}$ through the next linear layer $k - 1$ is in $\Omega(n_{k-1} n_{k-2}^2)$ as $\mathcal{H}_{k-2}$ has at least $n_{k-2}$ generators, with additional computations necessary to enclose the nonlinear layer $k$ (Prop. 3). As $\mathcal{O}(n_{k-1} n_{k-2}) \cap \Omega(n_{k-1} n_{k-2}^2) = \emptyset$, the computational overhead for the bound propagation is negligible. This becomes even more favorable with more complex set representations being used as they usually do not have a better computational complexity than zonotopes on the relevant operations.

**Theorem 1** (Sound Neural Network Reduction). *Given an input set $\mathcal{X}$, a neural network $\Phi$, and a reduction rate $\rho$, Alg. 2 constructs a reduced network $\widehat{\Phi}_\rho$ satisfying the problem statement in Sec. 2.5.*

*Proof.* For the main condition in the problem statement (Sec. 2.5)

$$\widehat{\Phi}_\rho(\mathcal{X}) \cap \mathcal{S} = \emptyset \implies \Phi(\mathcal{X}) \cap \mathcal{S} = \emptyset$$

to hold, we require that $\widehat{\Phi}_\rho(\mathcal{X}) \supseteq \Phi(\mathcal{X})$ holds. This is the case as each step in Alg. 2 is outer-approximative. In particular, as Prop. 4 holds, the computed error bounds ensure that the output of the original network is contained in the output of the reduced network. Please note that the subset relation holds for the exact evaluation of the respective sets, but might not hold if Prop. 3 is applied to compute the output set as a different abstraction might be used. $\square$

**Transformation of convolutional layers to linear layers.** Convolutional layers can be transformed to regular linear layers as defined in Def. 1 by viewing them as linear layers with shared weights (Bishop & Nasrabadi, 2006, Sec. 5.5.6). Let us briefly restate the definition of a convolutional layer for convenience:

**Definition 4** (Convolutional Layer (Bishop & Nasrabadi, 2006, Sec. 5.5.6)). *Given an input $I \in \mathbb{R}^{c_I \times h_I \times w_I}$ and a kernel $K \in \mathbb{R}^{c_O \times c_I \times h_K \times w_K}$, a convolutional layer computes the output $O \in \mathbb{R}^{c_O \times h_O \times w_O}$ for $k \in [c_O]$, $i \in [h_O]$, $j \in [w_O]$ as follows:*

$$O_{(k,i,j)} = \sum_{l=1}^{c_I} \sum_{m=1}^{h_K} \sum_{n=1}^{w_K} K_{(k,l,m,n)} I_{(l,i+m,j+n)},$$

*where $h_O = h_I - (h_K - 1)$, $w_O = w_I - (w_K - 1)$ and $c_I, c_O$ are the number of input and output channels, respectively.*

The same operation as in Def. 4 can be computed by flattening the input image $I$ into a vector:

$$\vec{I} = \begin{bmatrix} I_1 & \dots & I_{c_I} \end{bmatrix}^T,$$
$$\text{with } I_l = \begin{bmatrix} I_{(l,1,\cdot)} & \dots & I_{(l,h_I,\cdot)} \end{bmatrix}, \quad l \in [c_I], \tag{8}$$

and correctly populating each row of the weight matrix $W_K \in \mathbb{R}^{(c_O \cdot h_O \cdot w_O) \times (c_I \cdot h_I \cdot w_I)}$ with the kernel $K$:

$$W_K = \begin{bmatrix} K_{(1,1,1,1)} & K_{(1,1,1,2)} & \cdots & & 0 \\ 0 & K_{(1,1,1,1)} & \ddots & & 0 \\ \vdots & & \ddots & & \vdots \\ 0 & \cdots & & \cdots & K_{(c_O,c_I,h_K,w_K)} \end{bmatrix}. \tag{9}$$

Please note that $W_K$ is very sparse. If the convolutional layer has a bias, the construction of the resulting bias vector is analogous.

**Corollary 1** (Sound Compression). *Given an uncertain image $\mathcal{X} \subset \mathbb{R}_+^{c_I \times h_I \times w_I}$ and a reduction rate $\rho \in [0, 1]$, we can construct a neural network $\widehat{\Phi}_\rho$ that compresses this image with formal error bounds as follows: Let $K \in \mathbb{R}^{c_I \times c_I \times 1 \times 1}$ be a kernel of a convolutional layer, where*

$$K_{(k,l,1,1)} = \begin{cases} 1 & \text{for } k = l, \\ 0 & \text{otherwise,} \end{cases} \quad k, l \in [c_I], \tag{7}$$

Table 1: Network sizes and activation functions. Most networks are taken from VNN-COMP benchmarks (Brix et al., 2024)) and the others are variants from them provided by the team proposing the respective benchmark.

| Dataset | Benchmark | Activation | Description |
|---|---|---|---|
| MNIST | ERAN | ReLU | Fully-connected with $6 \times 100$ neurons |
| | | ReLU/Sigmoid | Fully-connected with $6 \times 200$ neurons |
| | | ReLU | Fully-connected with $6 \times 500$ neurons |
| | | ReLU/Sigmoid/Tanh | CNN with 2 convolutions and 2 linear layers |
| | | Sigmoid | Fully-connected with $6 \times 200$ neurons |
| | MNISTFC | ReLU | Fully-connected with $2 \times 256$ neurons |
| | | ReLU | Fully-connected with $4 \times 256$ neurons |
| | | ReLU | Fully-connected with $6 \times 256$ neurons |
| CIFAR-10 | Marabou | ReLU | Large CNN with 2 convolutions and 2 linear layers |
| | | ReLU | Medium CNN with 2 convolutions and 2 linear layers |
| | | ReLU | Small CNN with 2 convolutions and 2 linear layers |
| | Cifar2020 | ReLU | CNN with 4 convolutions and 3 linear layers |
| | ACAS-XU | ReLU | Fully-connected with $7 \times 50$ neurons |
| | QUAD | Sigmoid | Fully-connected with $3 \times 64$ neurons |

*and $\Phi$ be a neural network with two convolutional layers with kernel $K$ and one ReLU activation. The reduced network $\widehat{\Phi}_\rho$, obtained by applying Thm. 1 using $\mathcal{X}$, $\rho$, and dynamic merge buckets compresses the input $\mathcal{X}$ with sound error bounds according to $\rho$.*

*Proof.* The original network $\Phi$ computes the identity by construction. The image is compressed in the hidden layer of $\widehat{\Phi}_\rho$ (Thm. 1). The computed bounds ensure $\mathcal{X} \subseteq \widehat{\Phi}_\rho(\mathcal{X})$. □

**Corollary 2** (Reusing Reduced Network on Subsets)**.** *Given a neural network $\Phi$, an input set $\mathcal{X}$, and a reduction rate $\rho$, then a reduced network $\widehat{\Phi}_\rho$ according to Thm. 1 can be reused for all $\mathcal{X}' \subseteq \mathcal{X}$.*

*Proof.* The proof follows from the construction of $\widehat{\Phi}_\rho$ using Thm. 1. □

# B Additional Experiments and Ablation Studies

## B.1 Evaluation Details

All computations were performed on an Intel® Core™ Gen. 11 i7-11800H CPU @2.30GHz with 64GB memory if not stated otherwise. The GPU results for $\alpha, \beta$-CROWN were computed on the same device using a NVIDIA GeForce RTX 3080 laptop GPU. Details about the individual network can be found in Tab. 1. For all image datasets, we sample 100 correctly classified images from the test set and average the results. The perturbation radius $\epsilon \in \mathbb{R}_+$ is always stated with respect to the normalized images $\mathcal{X} \subset [0, 1]^{n_0}$. If not otherwise stated, feed-forward neural networks are reduced using static merge buckets and convolutional neural networks using dynamic merge buckets (Sec. 3.2).

**Figures showing remaining number of neurons:** For all figures showing the remaining number neurons while still verifying the given specification, our reduction results are illustrated by the mean remaining input neurons per nonlinear layer at even $k \in [\kappa]$. The number of input and output neurons of the entire network are displayed at $k = 1$ and $k = \kappa + 1$, respectively. We do not show the number of input neurons of a linear layer $k + 1$, as it has the same number of input neurons as the preceding nonlinear layer $k$. The number of neurons of the original network is shown in the same color with reduced opacity. Additionally, we show error bars indicating one standard deviation from the mean reduction per layer.

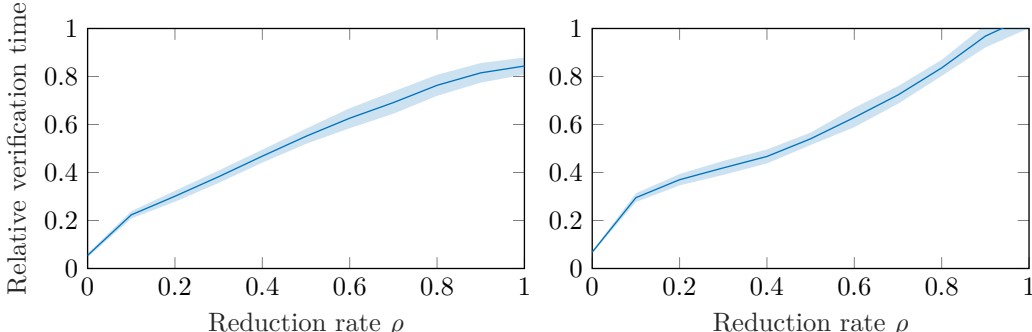

Figure 8: The relative verification time of the reduced network primarily depends on the reduction rate $\rho$: ERAN sigmoid network (left) and ERAN CNN (right).

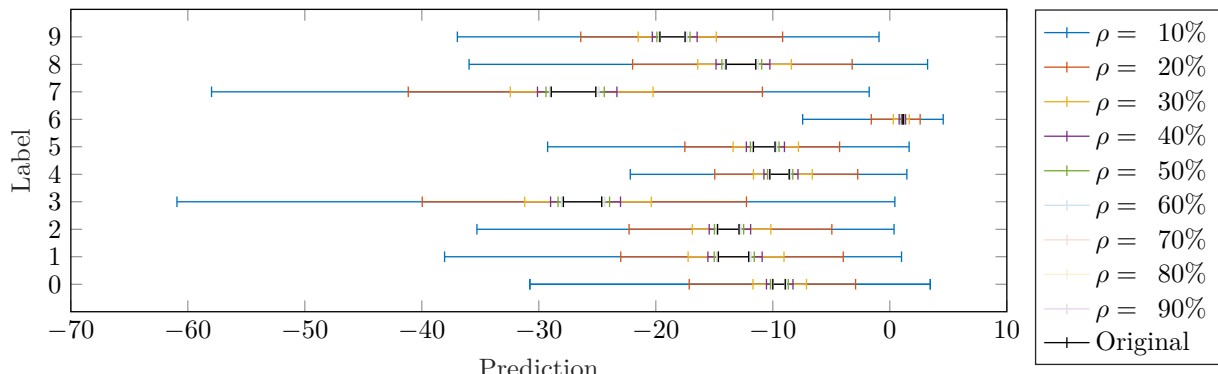

Figure 9: Fully automatic network reduction: Comparison of outer-approximative bounds of the prediction for an MNIST image with $\epsilon = 0.01$ using the ERAN sigmoid network for different reduction rates $\rho$.

## B.2 Ablation Study

In this section, we revisit all components and parameters to analyze their impact on the overall result.

### B.2.1 Varying the Reduction Rate $\rho$

Let us first analyze the effect of the reduction rate $\rho$ on the overall verification process. We exemplarily show the relative verification time of two networks in Fig. 8, where the shown times are normalized by the time to verify a single query using the original networks ($0.97s$ the for ERAN sigmoid network and $3.76s$ ERAN convolutional neural network). Please note that the times for $\rho < 1$ also include the time to reduce the network as the networks are reduced on-the-fly (Alg. 2). Thus, Fig. 8 shows that whenever the networks are reduced to a certain reduction rate $\rho$, the overall verification time is reduced to a similar degree.

For a challenging MNIST image with label 6, Fig. 9 shows the computed outer-approximative output bounds using the ERAN sigmoid network for different $\rho$. The bounds quickly converge with increasing $\rho$, and the image can be verified with $\rho \geq 30\%$ in this example.

### B.2.2 Varying the Bucket Tolerance $\delta$

Please note that our approach automatically tunes all internal parameters for any user-defined reduction rate $\rho$. In particular, the bucket tolerance $\delta$ is chosen accordingly such that the desired amount of neurons are merged (Alg. 1). To obtain a better understanding on this internal parameter, we vary the bucket tolerance $\delta$ for different perturbation radii $\epsilon$, where a larger value for $\delta$ results in a more aggressive merging, and show the corresponding reduction rate $\rho$ and number of verified instances in Tab. 2. The effects heavily depend on

Table 2: ERAN benchmark: Change of verification rate (VR) and reduction rate $\rho$ with varying perturbation radius $\epsilon$ and bucket tolerance $\delta$.

| Network $6 \times 200$ | | Sigmoid | | ReLU | |
|---|---|---|---|---|---|
| $\epsilon$ | $\delta$ | $\rho$ | VR [%] | $\rho$ | VR [%] |
| 0.0050 | 0.1000 | 0.2392 | 69.00 | 0.5496 | 76.00 |
| 0.0050 | 0.0100 | 0.3700 | 99.00 | 0.5500 | 100.00 |
| 0.0050 | 0.0050 | 0.4242 | 100.00 | 0.5573 | 100.00 |
| 0.0050 | 0.0010 | 0.5146 | 100.00 | 0.5607 | 100.00 |
| 0.0050 | 0.0001 | 0.6380 | 100.00 | 0.5618 | 100.00 |
| 0.0020 | 0.1000 | 0.1640 | 94.00 | 0.3602 | 95.00 |
| 0.0020 | 0.0100 | 0.3028 | 99.00 | 0.3556 | 100.00 |
| 0.0020 | 0.0050 | 0.3455 | 100.00 | 0.3549 | 100.00 |
| 0.0020 | 0.0010 | 0.4705 | 100.00 | 0.3479 | 100.00 |
| 0.0020 | 0.0001 | 0.6004 | 100.00 | 0.3494 | 100.00 |
| 0.0010 | 0.1000 | 0.1318 | 98.00 | 0.3143 | 92.00 |
| 0.0010 | 0.0100 | 0.2782 | 100.00 | 0.2846 | 100.00 |
| 0.0010 | 0.0050 | 0.3336 | 100.00 | 0.2931 | 100.00 |
| 0.0010 | 0.0010 | 0.4507 | 100.00 | 0.2909 | 100.00 |
| 0.0010 | 0.0001 | 0.5882 | 100.00 | 0.2882 | 100.00 |

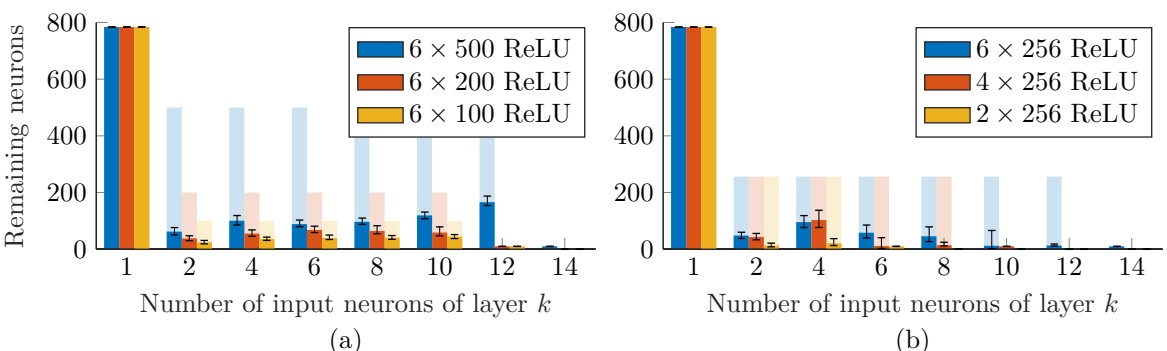

Figure 10: Reduction results with varying network sizes: (a) ERAN network variants and (b) MNISTFC networks.

the specifications and networks, such that the size of the resulting reduced network is not immediately clear. Fortunately, a user only has to specify the reduction rate $\rho$ and this internal parameter is tuned automatically.

### B.2.3   Varying the Network Size

Let us also analyze our approach with varying network sizes. In particular, we vary (i) the number of hidden neurons and (ii) the number of layers on an otherwise identical network architecture. For the first experiment, we take network variants from the ERAN benchmark[2], and show the result in Fig. 10a: Despite the different number of hidden neurons in the original network, the number of remaining neurons in the verified reduced network are not proportional to the respective original network. We assume that this is due to the typical over-parametrization of networks (Neyshabur et al., 2018), with larger networks having a larger over-parametrization, and our reduction approach extracts the relevant subnetwork. A similar observation

---

[2] ERAN network variants taken from the ERAN website: `https://github.com/eth-sri/eran`

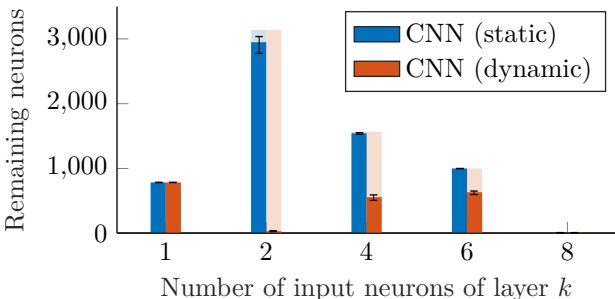

Figure 11: Convolutional neural networks require dynamic merge buckets to exploit similar neighboring pixels.

Table 3: Average reduction rates for large perturbation radii on the CORA benchmark. Entries marked with a hyphen ($-$) indicate combinations where not enough verifiable images were found for a meaningful average.

| Dataset | Training | $\epsilon = 0.01$ | $\epsilon = 0.02$ | $\epsilon = 0.05$ | $\epsilon = 0.1$ |
|---------|----------|-------------------|-------------------|-------------------|------------------|
| MNIST | Standard (Bishop & Nasrabadi, 2006) | 0.3338 | - | - | - |
| MNIST | Trades (Zhang et al., 2019) | 0.2760 | 0.3666 | - | - |
| MNIST | Set (Koller et al., 2024) | 0.1821 | 0.1879 | 0.2467 | 0.3556 |

can be made with increasing number of layers (Fig. 10b), where the networks are taken from the MNISTFC benchmark.

### B.2.4 Static vs. Dynamic Merge Buckets

Let us also analyze the types of merge buckets (Sec. 3.2). While saturated neurons are common in fully-connected networks (Fig. 2), these might not occur in convolutional neural networks. We show this in Fig. 11 on a convolutional neural network taken from the ERAN benchmark, where large reductions are only possible when dynamic merge buckets are used using a fixed bucket tolerance $\delta$.

### B.2.5 Large Perturbation Radii $\epsilon$

In this study, we analyze the impact of large perturbation radii on the network reduction while still verifying the specification. Please note that networks are generally not robust against large perturbation radii. In VNN-COMP (Brix et al., 2024), typical perturbation radii for standard neural networks are around $\epsilon = 0.01$ with respect to the normalized images $\mathcal{X} \subset [0,1]^{n_0}$. However, counterexamples can easily be extracted for larger radii and thus the specification is falsified. Of course, such networks can also not be verified using our approach. However, there is some literature on robustifying the networks (e.g., through adversarial training) and VNN-COMP also has a benchmark CORA featuring such networks, which can be used to demonstrate our approach on larger perturbation radii. We present the reduction results on those networks in Tab. 3.

### B.3 Additional Experiments

### B.3.1 Sound Image Compressing as Preprocessing Step

Fig. 12 shows how the sound compression preprocessing of the input image (Alg. 3) can further reduce the overall network size (ERAN CNN with sigmoid activation). The prepended layers shown at $-1$ and $0$ only have very few remaining neurons, where we only show the number of neurons corresponding to the dimension of the constructed zonotope. Thus, the average total verification time is reduced from 4.59s to 0.78s as the initial reduction (Alg. 3, Alg. 3) is computationally cheap and the representation of the involved sets is much smaller, i.e., the zonotopes have fewer generators, compared to verifying the original network. Our evaluation

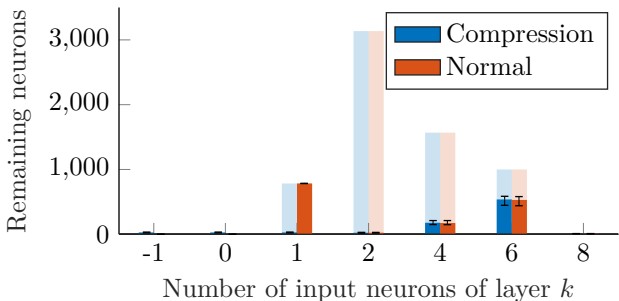

Figure 12: ERAN CNN with sigmoid activation: Input compression (Alg. 3) versus normal reduction (Alg. 2).

Table 4: Average network reduction and verification time of a property of the ACAS Xu benchmark.

| Neural Network | Number of Neurons | Verification Time |
|---|---|---|
| Original Network | 100.00 % | 7.38s |
| Reduced Network | 61.33 % | 4.21s |

shows that, on average, only 28 input neurons remain compared to the 784 input neurons of the original image (Fig. 12).

### B.3.2 Reusing a Reduced Network

Finally, we give two examples where the reduced network was reused despite its input set restriction (Cor. 2).

**ACAS Xu benchmark** We demonstrate the applicability of our approach on non-image data using the ACAS Xu benchmark (Bak et al., 2021). The benchmark consists of multiple networks and properties used to verify turn advisories to an aircraft to avoid collisions. The networks have $6 \times 50$ hidden layers with 5 input and 5 output neurons. As this benchmark is particularly hard to verify, we apply a branch-and-bound strategy by recursively splitting the input set along the most sensitive dimension. Using Cor. 2, we can reduce the network once on the original input set and reuse the reduced network to verify all subsets. The reduced network has on average only $\rho = 60\%$ of the neurons of the original network (Tab. 4, averaged over 10 runs). While Boudardara et al. (2023b) state that they can reduce the ACAS Xu networks down to a total number of 10 neurons, the obtained output sets are very conservative with a radius up to $10^{17}$ (Boudardara et al., 2023b, Fig. 14-16), which makes it impossible to verify the given specification.

**Closed-loop verification** Finally, let us revisit the quadrotor example from Sec. 4.2 to show the applicability of our approach in closed-loop systems: We enlarge the current reachable set at $t = 4s$ by a factor of 1.5 to compute the reduced network controller. This enlargement is necessary as the reachable set still oscillates around its equilibrium. The reduced network can then be reused in 60% of the remaining network evaluations. Whenever the current reachable set leaves the enlarged set used to reduce the network, the verification algorithm falls back to the original network until the reduced network can be used again. In the quadrotor example, the reduced network was always used again after one time step. Fig. 13 shows the relevant part of Fig. 5 and includes the reachable set computed with the reduced network, where both reachable sets remain within the desired goal region, and the reachable set computed with the reduced network is insignificantly larger.

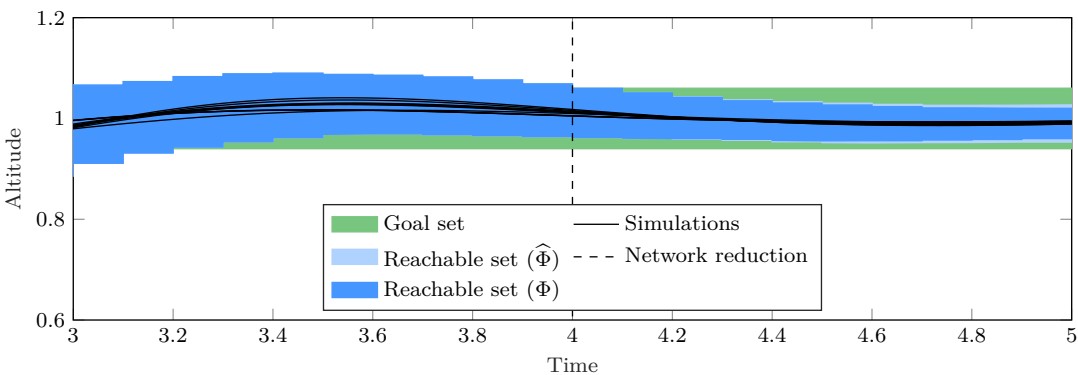

Figure 13: Quadrotor example: We can reuse a reduced network $\widehat{\Phi}$ once the system starts to converge to the desired altitude. The resulting reachable set is insignificantly larger.

