# OpenReview forum: "Fully Automatic Neural Network Reduction for Formal Verification"
_TMLR — Accepted by TMLR_

### Review · Reviewer_7ryJ · 2025-03-18

**Summary Of Contributions:**

This work observes that many neurons have similar activations in the specification and propose a new method to dynamically merge similar neurons to reduce the network size before running a formal verification. The proposed method is sound, meaning that verifications automatically transfer to the original unreduced network. The result shows that with the proposed method, even keeping an approximation network that has ~10% size is able to verify many inputs. This work concludes that the proposed method enjoys better timing than SOTA algorithms.

**Audience:**

Yes

**Broader Impact Concerns:**

No concern is identified.

**Claims And Evidence:**

Yes

**Requested Changes:**

1. While this is not necessary, I recommend the authors to re-implement their method in PyTorch/Tensorflow or any modern machine learning frameworks instead of MATLAB to allow others apply their methods. Note that all SOTA methods, e.g., $\alpha$-$\beta$-CROWN and MNBaB, have such implementations available.
2. Please include a study with a wider set of $\epsilon$, especially larger values.

**Strengths And Weaknesses:**

Strength:
1. the proposed method is sound, which is a good aspect for an approximation method;
2. the paper is well-written and well-motivated;
3. the experiments sufficiently support the claims.

Weakness:
1. the method is implemented in MATLAB, which greatly reduces its impact: SOTA algorithms are all implemented in PyTorch, allowing easy application and re-development;
2. the evaluated specification is small: $\epsilon=0.01$. I wonder if the approximation still works well for larger $\epsilon$, since a larger specification leads to more approximation errors in network compression.

---

> ### Author Response · Authors · 2025-05-09
> **Revision**
>
> Dear reviewer 7ryJ,
>
> Thank you for your time and valuable feedback.
>
> **Implementation in Python/PyTorch:**
> We agree that implementing our approach in a modern machine learning framework would be beneficial.
> We chose MATLAB/CORA for its rich set-based computing capabilities
> but we are in the process to also implement our approach in Python to maximize its impact.
> However, this will require some time as we also have to transfer the required capabilities of CORA to Python.
>
> **Larger Perturbation Radii:**
> We agree with the reviewer that an ablation study with larger perturbation radii is interesting.
> Please note that the perturbation radius cannot be increased arbitrarily as neural networks are usually not robust against larger radii
> and a counterexample can easily be found.
> Thus, the benchmarks in VNN-COMP usually also have a perturbation radius similar to $\epsilon=0.01$ with respect to the *normalized* image $\mathcal{X}\subset[0,1]^{n_0}$.
> However, there is some literature on robustifying the networks (e.g., through adversarial training)
> and VNN-COMP also has a benchmark featuring such networks.
> Following your suggestion, we have included an ablation study on this benchmark with larger perturbation radii in the paper (Sec. B.2.5).
>
> For your convenience, all changes are highlighted in blue.
>
> Best regards,
> The Authors

---

> > ### Comment · Reviewer_7ryJ · 2025-05-09
> >
> > Dear authors,
> >
> > Thanks for the reply. It clears my concerns.

---

### Review · Reviewer_HC6c · 2025-04-10

**Summary Of Contributions:**

The paper outlines an algorithm for the formal verification of neural networks, accommodating various activation functions such as ReLU, Sigmoid, and Tanh. The method builds upon an abstraction idea, where neurons within a neural network that exhibit similar behaviour regarding the property to be verified are merged. This results in a smaller network that can be verified in place of the original network without loosing soundness of the verification approach. The approach is referred to as "reduction" and can be used in combination with standard procedures such as zonotope- or intervall-based verfication approaches. The method's superior performance is demonstrated through experiments on standard benchmarks.

**Audience:**

Yes

**Broader Impact Concerns:**

There are none.

**Claims And Evidence:**

No

**Requested Changes:**

Somewhere, perhaps in the introduction, it should be made clearer what specific challenge this paper overcomes that related work did not. If I am not mistaken, the general approach is not entirely new; for example, refer to Ashok et al.'s "DeepAbstract: Neural Network Abstraction for Accelerating Verification". I am curious about what differentiates the setting in this paper from such related work. I noticed that this work is discussed in the related work section, but why isnt it, for example, compared to the methods this paper suggests  in the experiments?

Likewise, there must be some downside to the verification approach suggested here, compared to the non-reduction-based approaches of \alpha\beta-CROWN, no? Otherwise, I would be surprised why merging isnt used in all verifiers already, since the idea is around for a few years already. Assuming that I am not completely mistaken here, such drawbacks are not clear in the current paper.

**Strengths And Weaknesses:**

strengths:
- The paper is very polished and easy to understand.
- The main body of the paper is well-structured and straightforward to digest.
- The authors make an effort to provide a broad perspective.

weaknesses:
- It is not entirely clear what the specific contribution of this paper is. Although I am not an expert, neuron merging, pruning, or abstraction, or however it is termed, does not seem novel to me. Given that, I am surprised that such methods are not incorporated in existing high-competitive verifiers, especially considering the extreme superiority that the approach of this paper claims to have.

---

> ### Author Response · Authors · 2025-05-09
> **Revision**
>
> Dear reviwer HC6c,
>
> Thank you very much for your time and valuable feedback.
> We very much appreciate the acknowledgement of our paper writing,
> as clear and well-written papers are also very important to us.
>
> **Neural Network Reduction:**
> While there is a lot of literature on reducing the size of the network,
> these usually do not provide formal error bounds between the smaller network and the original network
> such that these cannot be applied to formal verification of neural networks.
> Thus, one has to look at sound neural network reduction approaches.
>
> **Sound Neural Network Reduction:**
> There exist a handful of sound neural network approaches as described in Sec. 1.1.
> These approaches generally analyze networks with specific activation functions (usually only ReLU)
> and do not consider any specifications during the reduction.
> In contrast, our approach works on all element-wise activation functions
> but the resulting reduced network is only valid on a specific input domain,
> i.e., as given by the specification (Sec. 2.5).
> This requires us to recompute the network reduction for different input domains.
> However, as the time to compute the reduced network is marginal compared to the time to verify the network (Fig. 8),
> this trade-off is favorable for us.
>
> **Comparisons to Other Sound Reduction Approaches:**
> As related work do not consider the specification during the network reduction,
> their reduction results are usually not very substantial or overly conservative:
>
> For example, $80-90$% of the neurons of the ERAN networks remain in related work once formal guarantees are demanded (Ashok et al., 2020, Fig. 2 & Tab. 2),
> whereas we reduce it to just $30$% for the ERAN networks (Fig. 7a) and $10-15$% for the Marabou networks (Fig. 7b) while still verifying the given specification.
> Additionally, our approach can handle networks with any type of element-wise activation function (Fig. 7c).
>
> While Boudardara et al. (2023b) state that they can reduce the ACAS Xu networks down to a total number of $10$ neurons,
> the obtained output sets are very conservative with a radius up to $10^{17}$ (Boudardara et al., 2023b, Fig. 14-16),
> which makes it impossible to verify the given specification.
>
> Furthermore, the networks considered by related work are usually much smaller than networks we considered.
> For example, the Cifar2020 benchmark (Fig. 7d) has more neurons in a single layer than related work consider in total (Boudardara et al., 2023c).
>
> **Integration in SOTA Verifiers:**
> We believe that the points mentioned above limited the usage of network reduction in SOTA verifiers
> and with our approach, network reduction will also be adopted by them.
> Please note that this can easily be done for approaches based on reachability analysis (see discussion in Sec. 2.3).
> For optimization-based approaches, it might be harder to integrate
> but, e.g., $\alpha-\beta$-CROWN also runs an initial bound propagation using the auto-LiRPA library,
> which can probably be used as bounds to reduce the network as well.
>
> We added these comparisons and clarifications to the paper.
> For your convenience, all changes are highlighted in blue.
>
> Best regards,
> The Authors

---

> > ### Comment · Reviewer_HC6c · 2025-05-19
> >
> > Thanks to the authors, this answers my questions.

---

### Review · Reviewer_1aVQ · 2025-05-08

**Summary Of Contributions:**

The paper proposes a novel and fully automatic technique for the sound reduction of neural networks to enhance the scalability of formal verification tools. The core idea is to reduce the number of neurons in a neural network while preserving soundness, meaning that any verification result on the reduced network guarantees the same result for the original network. The method applies to networks with element-wise activation functions (ReLU, sigmoid, tanh) and integrates reachability analysis using zonotopes to ensure soundness. The technique dynamically merges similar neurons on-the-fly, which avoids constructing high-dimensional sets during verification, thereby improving efficiency.

**Audience:**

Yes

**Broader Impact Concerns:**

No concerns

**Claims And Evidence:**

Yes

**Requested Changes:**

The paper is well-written and clear. Despite what I reported as weaknesses, the paper is a good contribution that, in my opinion, does not require significant work.

I list some suggestions:
- in the introduction the content of the paper is only summarized with bullet points but not discussed in detail for example with respect to related works. I would suggest improving the introduction to better frame what the paper proposes with respect to competitors.
- in section 2.2 the description of neural networks is not clear to me, especially in the differences between linear and nonlinear layers. For example, the paper always talk about "linear and nonlinear" but maybe talking about "linear or nonlinear" layers would be more correct. I suggest rewriting this section.
- algorithm 1 could be written using the entire line width instead of half page, this will improve readability of comments and pseudocode.

**Strengths And Weaknesses:**

Strengths
- significant speed-up (up to 96%) in verification demonstrated across multiple benchmarks (ERAN, Marabou, CIFAR-10).
- applicable to convolutional networks.
- significantly reduces computation even on CPU-only systems, which is essential for embedded systems.

Weaknesses
- the merging is based on bounds derived from interval arithmetic during look-ahead steps. While efficient, this could be overly conservative in high-dimensional or highly nonlinear settings, potentially reducing effectiveness.
- while the method is extended to convolutional networks, recurrent or transformer-based architectures are not considered, potentially limiting the scope in certain domains like NLP.

---

> ### Author Response · Authors · 2025-05-09
> **Revision**
>
> Dear reviewer 1aVQ,
>
> Thank you very much for your time and valuable feedback.
>
> **Weaknesses:**
>
> **Look-ahead with interval arithmetic:**
> The reviewer is absolutely right in their assessment that interval arithmetic can lead to very conservative results.
> We want to stress that this is only applied for one step
> to reduce the outer approximation induced by the wrapping effect.
> Additionally, we only merge the neurons with the smallest bounds (Fig. 2, Def. 3);
> thus, only merging neurons for which this effect did not have a large influence.
>
> **Extension to other architectures:**
> In recent years, the formal neural network verification community started to explore other architectures [1, 2, 3] than standard feed-forward architectures.
> We agree that the extension of our network reduction to such architectures is a valuable future direction of our work.
>
> **Requested changes:**
>
> **Improved related work comparison:**
> We made some changes in the related work section leading up to the contributions as well as in the contributions itself.
> We hope this clarifies what our paper proposes compared to the competitors.
>
> **On linear and/or nonlinear layers:**
> Some works formulate the layers of a neural network as $L_k(h_{k-1})=\phi_k(W_k h_{k-1} + b_k)$,
> i.e., having the linear part and the nonlinear part within one layer.
> We opted to separate them into individual layers as the enclosure is handled differently,
> however, the architecture is the same.
> Thus, we always have linear *and* nonlinear layers in alternating fashion.
>
> **Half-page algorithms:**
> We agree with the reviewer that the algorithms, in particular Alg. 1,
> were unnecessarily compacted into one half of a page.
> To make them more readable, we now have them span over the entire line width.
>
> We added these changes and clarifications to the paper.
> For your convenience, all changes are highlighted in blue.
>
> Best regards,
> The Authors
>
> References:
>
> [1] Bonaert et al. "Fast and precise certification of transformers." SIGPLAN International Conference on Programming Language Design and Implementation. 2021.
>
> [2] Ryou et al. "Scalable polyhedral verification of recurrent neural networks." CAV. 2021.
>
> [3] Ladner et al. "Formal Verification of Graph Convolutional Networks with Uncertain Node Features and Uncertain Graph Structure." TMLR. 2025.

---

> > ### Comment · Reviewer_1aVQ · 2025-05-12
> > **Regarding linear and nonlinear layers**
> >
> > Now it is clear. I would suggest reporting your comment in the paper. I think it is still not straightforward to follow this part of the paper.

---

> > > ### Author Response · Authors · 2025-05-12
> > >
> > > Glad to hear. We updated the pdf and included this paragraph, as suggested. Thank you again for your feedback.

---

### Author Response · Authors · 2025-06-20
**Rebuttal Summary**

Dear Action Editor, Dear Reviewers,

Thank you again for organizing the review and the valuable feedback.
Below, we briefly summarize the rebuttal phase.

- We clarified some paragraphs and adjusted the formatting of algorithms that reviewers flagged as unclear to improve the readability of our paper.
- We added comparisons to related work in the evaluation section to clearly show the advantages of our work over them.
- We added ablation studies with large perturbation radii applied to adversarially trained networks

While the initial feedback from all reviewers was already quite positive, all reviewers commented that their open questions were answered and any concerns were cleared.

Best regards,
The Authors.

---

### Decision · Action_Editor_rDXd · 2025-06-20

**Recommendation:** Accept as is

**Additional Comments:**

This paper studies formal verification of neural networks. The authors introduce a novel, fully automated method for sound reduction of neural networks, aimed at improving the scalability of formal verification tools. The proposed approach involves reducing the number of neurons in the network while ensuring soundness—i.e., any verification outcome obtained on the reduced model remains valid for the original network.

All reviewers agree that the proposed approach is sound, the manuscript is well-written, and the experimental results are compelling. The reviewers raised several concerns, all of which were addressed by the authors during the discussion phase. Therefore, I recommend accepting this submission.

I strongly encourage the authors to incorporate the reviewers’ comments and suggestions regarding clarifications and additional details, doing so will significantly strengthen the paper.

**Audience:**

Yes

**Audience Explanation:**

This paper deals with formal verification with neural network. I believe this would be relevant for the ML community.

**Claims And Evidence:**

Yes

**Claims Explanation:**

All reviewers agree that the proposed method is sound, experimental results are reported properly, and claims made in the paper has supported evidence.

---

> ### Author Response · Authors · 2025-06-26
> **Camera-Ready Version**
>
> Dear all,
>
> Thank you very much for organizing the review and all the valuable feedback. We just uploaded the camera-ready version, where we included all suggested changes and additional experiments.
>
> Best regards, the Authors.